# Testing the Münch hypothesis of long distance phloem transport in plants

Michael Knoblauch[1]*, Jan Knoblauch[1,2], Daniel L Mullendore[1], Jessica A Savage[2,3], Benjamin A Babst[4†], Sierra D Beecher[1], Adam C Dodgen[1], Kaare H Jensen[5], N Michele Holbrook[2]

[1]School of Biological Sciences, Washington State University, Pullman, United States; [2]Department of Organismic and Evolutionary Biology, Harvard University, Cambridge, United States; [3]Arnold Arboretum of Harvard University, Boston, United States; [4]Department of Biosciences, Brookhaven National Laboratory, Upton, New York; [5]Department of Physics, Technical University of Denmark, Lyngby, Denmark

**Abstract** Long distance transport in plants occurs in sieve tubes of the phloem. The pressure flow hypothesis introduced by Ernst Münch in 1930 describes a mechanism of osmotically generated pressure differentials that are supposed to drive the movement of sugars and other solutes in the phloem, but this hypothesis has long faced major challenges. The key issue is whether the conductance of sieve tubes, including sieve plate pores, is sufficient to allow pressure flow. We show that with increasing distance between source and sink, sieve tube conductivity and turgor increases dramatically in *Ipomoea nil*. Our results provide strong support for the Münch hypothesis, while providing new tools for the investigation of one of the least understood plant tissues.

*For correspondence: knoblauch@wsu.edu

Present address: †School of Forestry and Natural Resources, University of Arkansas at Monticello, Arkansas, United States

Competing interests: The authors declare that no competing interests exist.

## Introduction

Vascular systems allow organisms to distribute resources internally by bulk flow and thus to overcome size limitations set by diffusion. In plants, the evolution of vascular tissues enabled the development of trees and forests and was accompanied by a major increase in the productivity of terrestrial ecosystems. Plant vascular tissues are of two types: the xylem allows water to be pulled from the soil to maintain the hydration of leaves surrounded by air, while the phloem distributes the products of photosynthesis throughout the plant, allowing non-photosynthetic structures, such as roots, to be formed. The current hypothesis for phloem transport dates to 1930 when Ernst Münch proposed that transport through the phloem results from osmotically generated differences between the pressure in sources (e.g. leaves) and sinks (e.g. roots), and occurs without any additional input of energy along the transport path (*Münch, 1930*). The Münch hypothesis has gained wide acceptance based to a large extent on its simplicity and plausibility, rather than on experimental evidence.

A fundamental issue bedeviling the pressure flow hypothesis is the long standing question of whether sieve tubes have sufficient hydraulic conductivity to allow sugars to be transported from leaves to roots in the largest and longest of plants. Increasing distances between sources and sinks would appear to require greater pressure to drive flow, but measurements suggest lower source turgor in trees compared to smaller, herbaceous plants (e.g., crops or weeds; *Turgeon, 2010*). A number of authors have emphasized the difficulty of accommodating the Münch hypothesis in trees, leading to the proposal of re-loading mechanisms via relays (*Lang, 1979*; *Aikman, 1980*). However, it is important to note that such challenges derive in large part from mathematical models of phloem

**eLife digest** Plants use energy from sunlight to make sugars in a process called photosynthesis. Most photosynthesis takes place in the leaves and so much of the sugar needs to be transported to other parts of the plant, such as fruits or roots. The sugars are transported by phloem tubes, which form a system that spans the entire plant.

In 1930, a German scientist called Ernst Münch proposed a hypothesis for how phloem tubes move sugars and other molecules around the plant. He proposed that the loading of these molecules into phloem tubes in the leaves or other "source" tissues makes the fluid inside the vessels more concentrated so that water is drawn into the phloem from neighboring "xylem" vessels. This creates pressure that pushes the fluid along the phloem tube towards the fruit, roots and other "sink" tissues. In the sink tissues the sugars are consumed, which reduces their concentration in the phloem and the pressure. Overall, this results in the flow of sugars and other molecules from where they are produced to where they are most needed.

However, this hypothesis is still largely untested because it has proved difficult to carry out experiments on phloem. Detaching the source tissues from the sink tissues stops the flow of fluid so only experiments in whole plants can provide meaningful data. Knoblauch et al. have now developed new methods to study phloem in an ornamental plant called morning glory.

The experiments show that plants can alter the shape of phloem vessels and the pressure within the vessels to allow them to transport sugars and other molecules over different distances. These findings strongly support the Münch hypothesis and make other alternative hypotheses seem unlikely. Furthermore, the methods developed by Knoblauch et al. will allow others to further investigate phloem transport. New findings in this area may allow plant biologists to direct the flow of sugars and other molecules towards particular plant tissues to improve the nutritional quality of food crops in the future.

transport in which the values of one or more of the key variables are unknown or poorly constrained (*Tyree et al., 1974*; *Pickard and Abraham-Shrauner, 2009*; *Thompson and Holbrook, 2003*; *Höltta et al., 2009*).

Phloem transport ceases immediately when sources and sinks become disconnected, which is always the case when preparing sieve tubes for in vitro studies. *In situ* studies on the other hand are challenging, especially in large plants. Thus, despite its importance, phloem transport and resource allocation is the least understood major process in plant function. Over the last years we have developed *in situ* methods to measure with high precision all of the parameters needed to quantify laminar flow through passive microtubes as described by the Hagen-Poiseuille equation

$$U = \frac{k}{\eta}\frac{\Delta p}{L}, ,\tag{1}$$

where $U$ is the flow velocity, $\Delta p$ is the pressure differential, $L$ is the tube length, $\eta$ is the sap viscosity, and $k$ is the conductivity of the tube. Previously, methods only existed for two of the five parameters: length, which is the distance between source and sink, and velocity, which can be determined using radioisotopes (*Babst et al., 2005*), magnetic resonance imaging (*Mullendore et al., 2010*), or dye tracking (*Savage et al., 2013*). We have recently introduced methods to measure the missing parameters including new EM preparation protocols and mathematical modeling to quantify sieve tube hydraulic resistance (*Froelich et al., 2011*; *Mullendore et al., 2010*; *Jensen et al., 2012*); a novel micro-capillary pressure probe to determine pressure in small and sensitive cells such as sieve tubes (*Knoblauch et al., 2014*); and protocols for *in situ* observation and flow velocity measurements in individual tubes (*Froelich et al., 2011*). A new method for determining phloem sap viscosity in vivo using fluorescence lifetime imaging of the fluorophore 2-NBDG is described.

Here we present a study on phloem flow relevant parameters that tackles the major question on the pressure flow hypothesis. Our data provide strong support for the Münch hypothesis as a unifying mechanism of long distance transport in plants. The results also call into question current understanding of downstream events such as the "high pressure manifold model" of phloem unloading

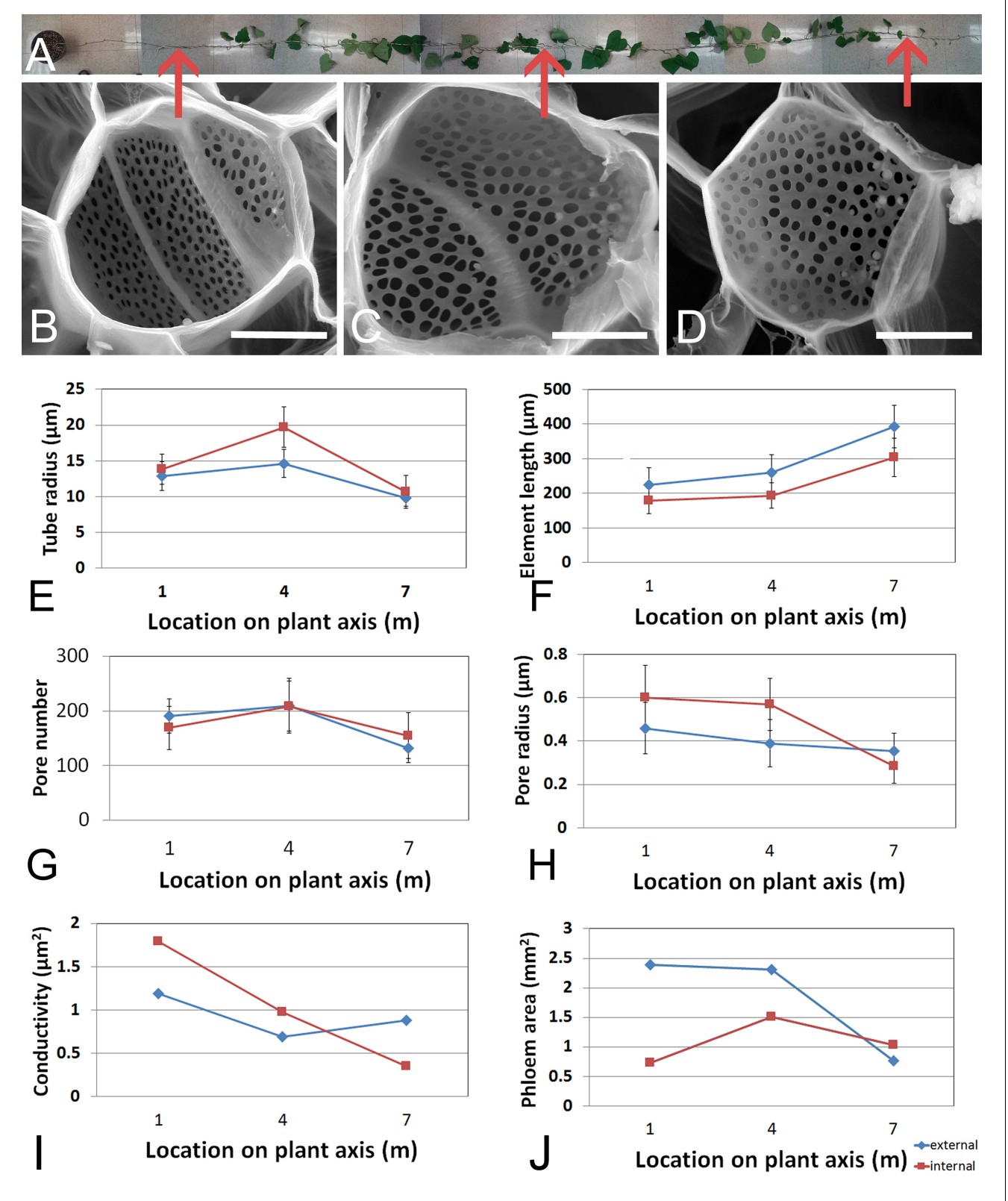

**Figure 1.** Geometrical parameters of a moderate sized morning glory plant. (**A**) A 7.5 m long morning glory plant with source leaves along the stem. (**B–D**) Scanning electron micrographs of sieve plates at 1 m (**B**), 4 m (**C**), and 7 m (**D**) from the base of the stem (as indicated by red arrows in **B–D**). **E–J**) Cell geometrical data were collected separately for internal (red) and external (blue) phloem. Average sieve element radius (**E**; n ≥ 10 per data point),
*Figure 1 continued on next page*

*Figure 1 continued*

length (**F**; n ≥ 10 per data point), pore number (**G**; n ≥ 10 per data point), and pore radius (**H**; n > 150 per data point) result in a conductivity of ~1 µm$^2$ for the external phloem but an increasing conductivity along the stem for the internal phloem (**I**). The total phloem area (**J**) is, however, significantly higher for the external phloem. Error bars show standard deviation. Scale bars in **B–D** =10 µm

The following source data and figure supplement are available for figure 1:

**Source data 1.** Source data of sieve tube geometrical parameters for *Figure 1* and *Figure 3—figure supplement 3*.

**Figure supplement 1.** Stem anatomy of morning glory plants.

and carbohydrate delivery to sinks. The toolset described here will allow for detailed research on phloem transport and resource allocation, processes critical for food security and improvement of bioenergy crops, as well as understanding ecosystem ecology and the global carbon cycle.

## Results

To test whether phloem parameters scale in accord with the Münch hypothesis, we chose to study morning glory (*Ipomoea nil*) vines because this species exhibits indeterminate growth, thus allowing us to test the predictions of the Münch model for a wide range of tube lengths (*L*). Moreover, its growth pattern is easy to manipulate, and it loads various fluorophores into its phloem. To simplify analysis, we allowed the plants to produce only one axis with two major sinks (root and shoot tip) by pruning all developing side branches and flowers daily.

### Sieve tube specific conductivity (*k*)

We first acquired baseline parameters from 7.5 m long plants (*Figure 1*) and measured anatomical parameters at locations 1 m, 4 m, and 7 m (*Figure 1 E-J*). The acquisition of geometrical parameters requires the highest accuracy because minor changes in tube geometry have a large impact on the calculated conductivity. This is the main reason why current models have to cope with large uncertainties. In addition, the natural variation of tube parameters requires the acquisition of large data sets. For this study, more than 1000 SEM and confocal images were taken, >100,000 sieve plate pore numbers were counted (in ≥10 plates per data point), and >15,000 sieve plate pore diameters (n ≥ 380 per data point), >1500 sieve element diameters (n ≥ 10 per data point), and >800 sieve element lengths (n ≥ 10 per data point) were measured. Changes in sieve tube geometry including sieve element radius (*Figure 1E*), sieve element length (*Figure 1F*), sieve plate pore number (*Figure 1G*) and sieve plate pore radius (*Figure 1H*) caused a slight increase in sieve tube specific conductivity (see appendix for specifics on conductivity calculations) towards the base of the 7.5 m plant (*Figure 1I*) of around 1 µm$^2$. Measurement of the total phloem area from high resolution confocal images of stem cross sections (*Figure 1—figure supplement 1A*) showed similar external and internal phloem areas during primary growth (*Figure 1—figure supplement 1B*), but a much larger increase of external phloem area during secondary growth.

### Sieve tube sap flow velocity (*U*)

The basipetal flow velocity on the stem was determined by application of $^{11}CO_2$ to one source leaf approximately 3 m from the shoot base (see Materials and methods section). The average flow velocity was 123 ± 13 µm/s (n = 3; *Figure 2*).

### Sieve tube sap viscosity (η)

Currently, phloem sap viscosity has not been directly measured, but only estimated based on the concentrations of extracted phloem sap contents (*Thompson and Holbrook, 2003*). Recently, a group of dyes has been identified that can be loaded into the phloem (*Knoblauch et al., 2015*). We tested carboxyfluorscein diacetate (CFDA), 8-Hydroxypyrene-1,3,6-trisulfonic acid acetate (HPTSA), Esculin, 2-(*N*-(7-Nitrobenz-2-oxa-1,3-diazol-4-yl)Amino)-2-Deoxyglucose (2-NDBG), and carboxytetraethylrhodamine glucoside (CTER) for potential molecular rotor properties which allow the measurement of solute viscosity due to viscosity dependent changes in fluorescence decay times

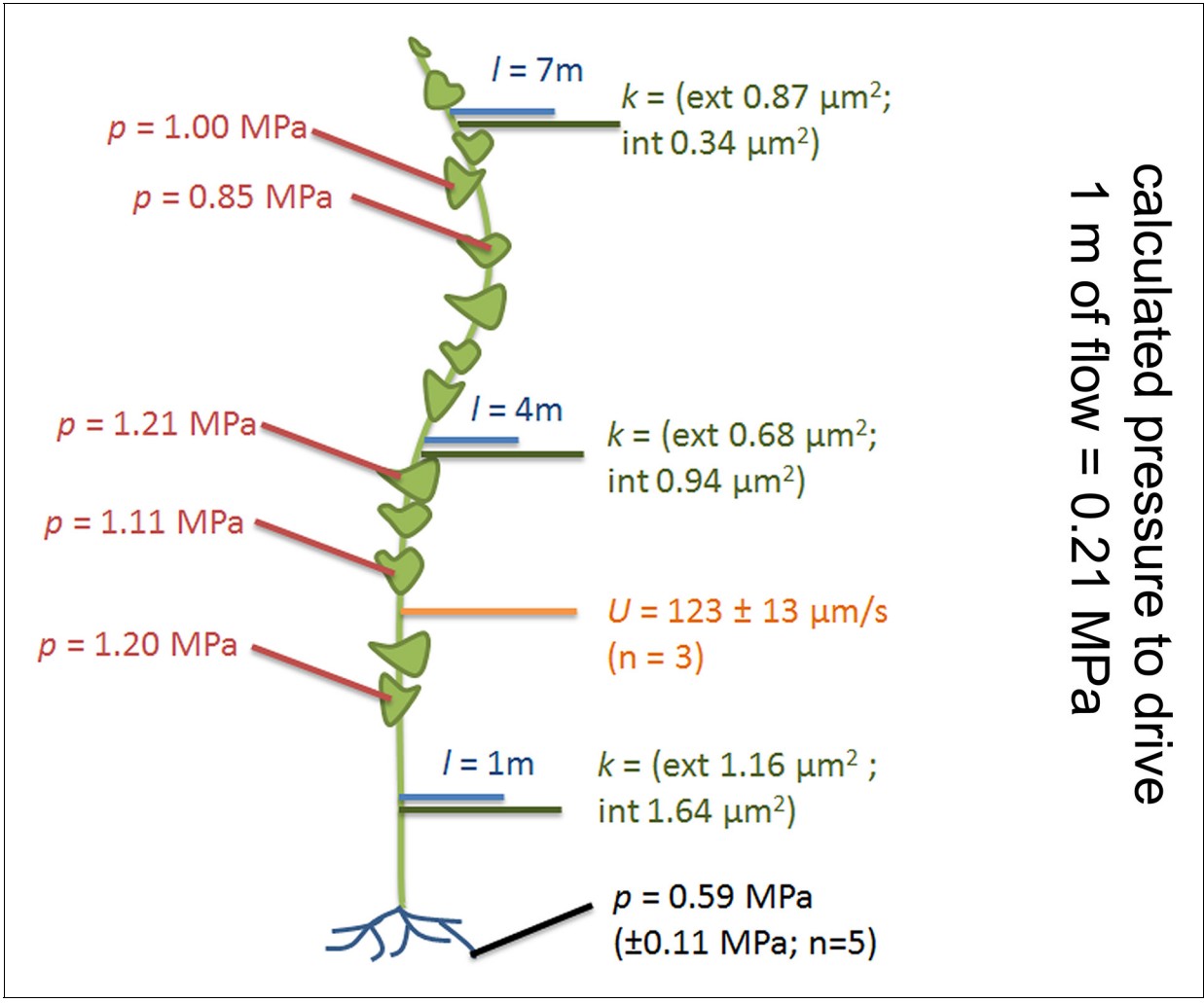

**Figure 2.** Phloem flow relevant parameters in a medium sized morning glory plant. An illustration summarizing the findings in a medium sized plant with leaves attached along the entire length of the stem. Cell geometrical data were taken at 1 m, 4 m, and 7 m (blue lines) along the stem. Resulting conductivities are indicated in green. Source sieve tube turgor measurements (red) were taken in the main vein of leaves along the stem axis, sink turgor (black) in root tips, and average flow velocity was measured using $^{11}CO_2$ labeling (orange).

The following figure supplements are available for figure 2:

**Figure supplement 1.** *In situ* viscosity measurements.

**Figure supplement 2.** *In situ* sieve tube turgor measurements.

**Figure supplement 3.** Symplastic phloem unloading in the root tip of morning glory plants.

determined with fluorescence lifetime imaging (FLIM). FLIM calibrations confirmed that 2-NBDG is a molecular rotor. To determine the phloem sap viscosity *in situ*, sieve elements were loaded with 2-NBDG and the decay of fluorescence lifetime was measured to generate a color coded viscosity map of the phloem (*Figure 2—figure supplement 1*). The average fluorescence lifetime of 2-NBDG in source phloem in morning glory was 1.366 ns (± 0.037 ns, n = 8), which corresponds to a viscosity of 1.7 mPas or a sucrose concentration of about 18%. This value is within the range of estimates based on solute concentrations in phloem sap extracts of various plant species (*Jensen et al., 2013*).

Using these parameters (σ,= 1.7 mPas, k = 1 μm², U = 123 μm/s) we estimated that a pressure differential of 0.21 MPa would be required to drive flow through a 1 m long tube of the determined anatomy. To evaluate if such a pressure differential exists between source and sink tissue, we measured phloem pressure *in situ*.

## Sieve tube turgor pressure (*p*)

Because the phloem is embedded in a thick tissue layer, *in situ* experiments have to be carried out by removing cortical tissue until the sieve tubes are exposed, but care must be taken to not injure the tubes. The preparation was achieved by making cortical hand sections with fresh razor blades as described earlier (*Knoblauch and van Bel, 1998*). Other methods such as laser ablation or sectioning by micromanipulators appear more elegant, but turned out to cause massive artifacts and flow stoppage. The section has to be as accurate as a twentieth of a millimeter in order to be useful for measurements of sieve tubes. This reduces the success rate to below 30% even for very experienced investigators. If sieve tubes were injured, the leaf was removed and the plant was allowed to recover for at least 24 hr. If a section appeared successful, upstream application of phloem mobile fluorescent dyes was used to verify intactness by monitoring translocation of the dye through the tube (*Video 1*, *Figure 2—figure supplement 2*). The preparation required the tissue to be covered with an artificial medium, which could have had an influence on sieve tube turgor. However, the low resistance between sieve elements provided a buffer for local disturbance. If the medium resulted in a local increase in turgor, an artificial source would have been generated and flow would reverse towards the leaf and no fluorochromes would have arrived at the site of observation. If preparation decreased turgor, a local sink would have been induced and flow would have been towards the area of preparation from both sides. The applied fluorochromes would eventually have arrived at the site of observation but would not have passed by. In all cases, turgor measurements were conducted on tubes showing quickly increasing fluorescence downstream of the site of preparation indicating that the applied medium did not disturb turgor pressure inside the sieve tube.

Sieve tube turgor in source leaves (*Figure 2*) at different levels along the stem averaged 1.08 MPa (± 0.13 MPa, n = 5). The anatomy of sinks (e.g. roots) does not permit direct measurements in sieve tubes. However, since phloem unloading in root tips follows a symplastic path (*Figure 2—figure supplement 3*; *Wright and Oparka, 1997*; *Patrick, 1997*; *2013*) the pressure in root cortical cells cannot be (significantly) higher than in sink sieve tubes. In situ turgor measurements in the cortex of the root elongation zone revealed a sink turgor of 0.59 MPa (± 0.11 MPa; n = 5). Therefore the measured pressure differential of 0.49 MPa could account for a pressure flow since the distance from a sink to the closest source did not exceed 2 m and a pressure gradient of 0.21 MPa/m is required to drive the flow.

## Manipulation of transport length (*l*) to monitor if flow relevant parameters scale according to pressure flow requirements

Having established the baseline parameters in a moderate-sized plant, we investigated if the parameters scale with increasing transport distance, a treatment we achieved by increasing the length of the stem without source leaves (*Figure 3*; *Figure 3—figure supplement 1*). In response to this treatment, the plant must change conductivity, pressure, velocity, or viscosity (or any combination of those) to maintain sink assimilate delivery if a passive pressure flow is the mode of translocation. Over the course of 5 months all newly developing tissues (leaves, side shoots) below the top 4 m were pruned daily. Phloem pressure was measured in situ (*Video 1*)

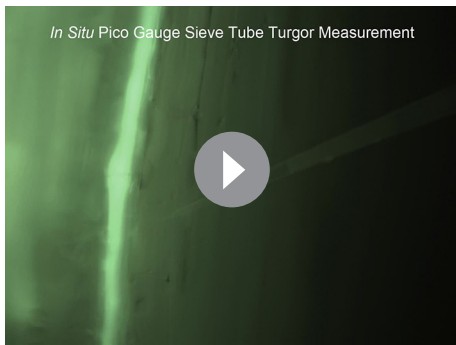

In Situ Pico Gauge Sieve Tube Turgor Measurement

**Video 1.** *In situ* sieve tube turgor pressure measurement using a pico gauge. Real time video of a pico gauge measurement. The fluorescence of a distantly loaded sieve tube provides evidence for tube intactness and transport. The oil – water interface in the pico gauge moves backwards when the pico gauge is impaled into the sieve tube, indicating compression of the oil volume. Inflow of fluorescent dye into the pico gauge tip provides evidence that the measurement occured in the sieve tube.

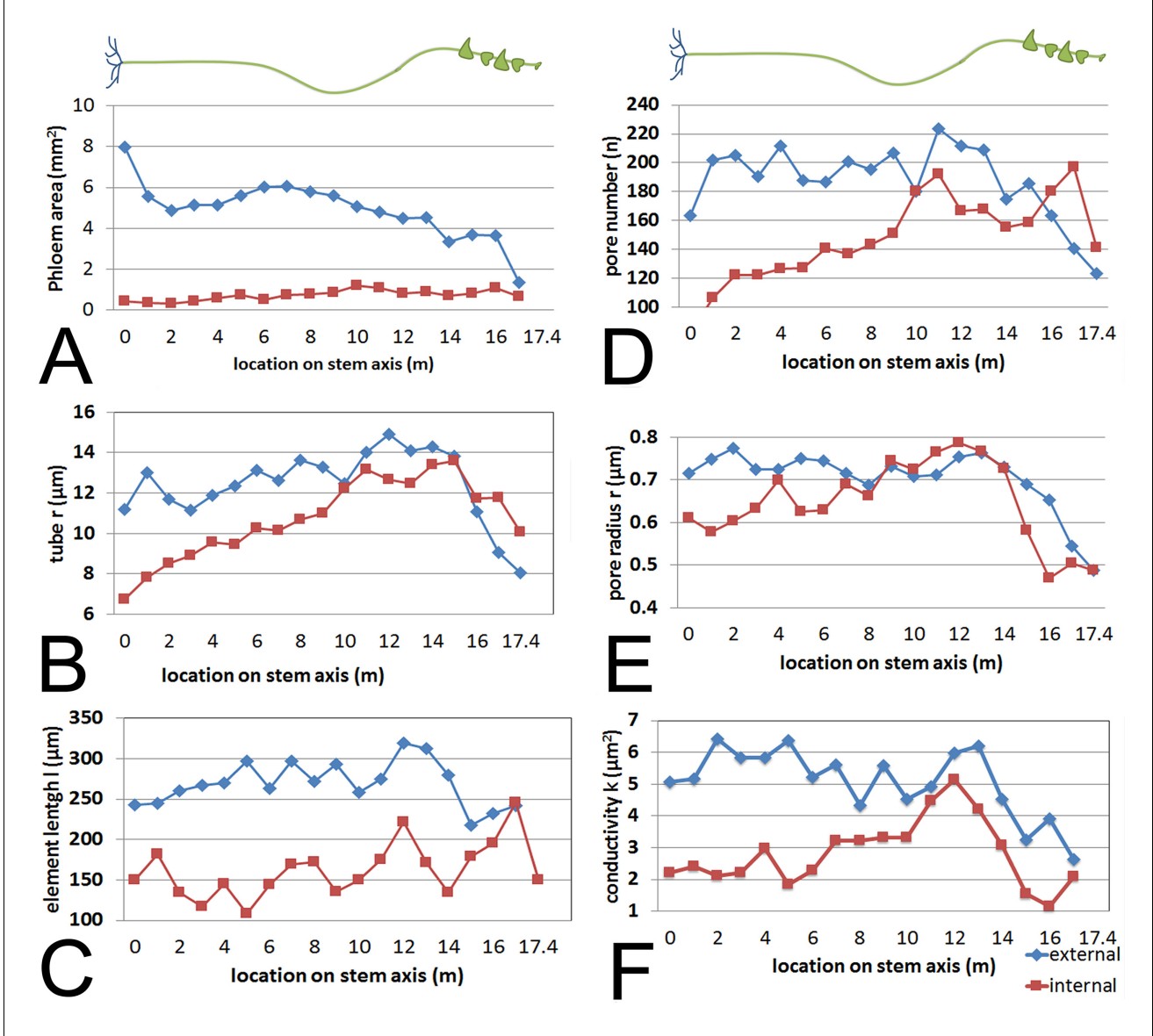

**Figure 3.** Geometrical parameters of a large morning glory plant with partially defoliated stem. Geometrical data of a 17.5 m long morning glory plant after 5 months growth with daily removal of developing side branches and flowers as well as removal of source leaves below the top 4 m. (**A**) Total phloem area at different locations along the shoot. Plotting phloem area versus distance indicates that only the external phloem (blue) increases in area significantly (internal phloem, red). (**B–F**) Cell geometrical data for sieve element radius (**B**; n ≥ 10 per data point), sieve element length (**C**; n ≥ 10 per data point), sieve plate pore number (**D**; n ≥ 10 per data point), and sieve plate pore radius (**E**; n ≥ 380 per data point) reveal that sieve tube conductivity (**F**) increases with the length of the transport pathway. Please see *Figure 3—figure supplement 3* for a comparison of the parameters and standard deviations between the moderate sized foliated morning glory (*Figure 1*) and the partially defoliated large morning glory plant.

The following source data and figure supplements are available for figure 3:

**Source data 1.** Source data of sieve tube geometrical parameters for *Figure 3* and *Figure 3—figure supplement 3*.

**Figure supplement 1.** Habitus and anatomy of a partially defoliated large morning glory plant.

**Figure supplement 2.** Anatomical adaptation to increase sieve tube conductivity.

**Figure supplement 3.** Comparison of geometrical parameters between a small foliated (compare *Figure 1*; green = external phloem, black = internal phloem) and a large, partially defoliated (*Figure 3*; blue = external, red = internal) morning glory plant.

in the main vein of the lowermost leaf throughout the growth period until the single stem of the plant was 17.5 m long with about 14 m of stem free of source leaves (*Figure 4*). At this time point, we measured phloem flow velocity by $^{11}CO_2$ application and used micro PET scanning to confirm phloem translocation direction (*Figure 4—figure supplement 1*). Afterwards, the tissue was harvested, cell geometrical data were collected at 1 m increments along the stem, and tube conductivity was estimated (*Figure 3 A-F*).

Source sieve tube turgor pressure was 0.75 (± 0.05 n = 3) MPa in young plants and increased to more than 2.2 MPa at the end of the experiment (*Figure 4*) when the distance from the lowest leaf to the base of the stem was 14 m with a transport distance of 17.5 m when applying a correction

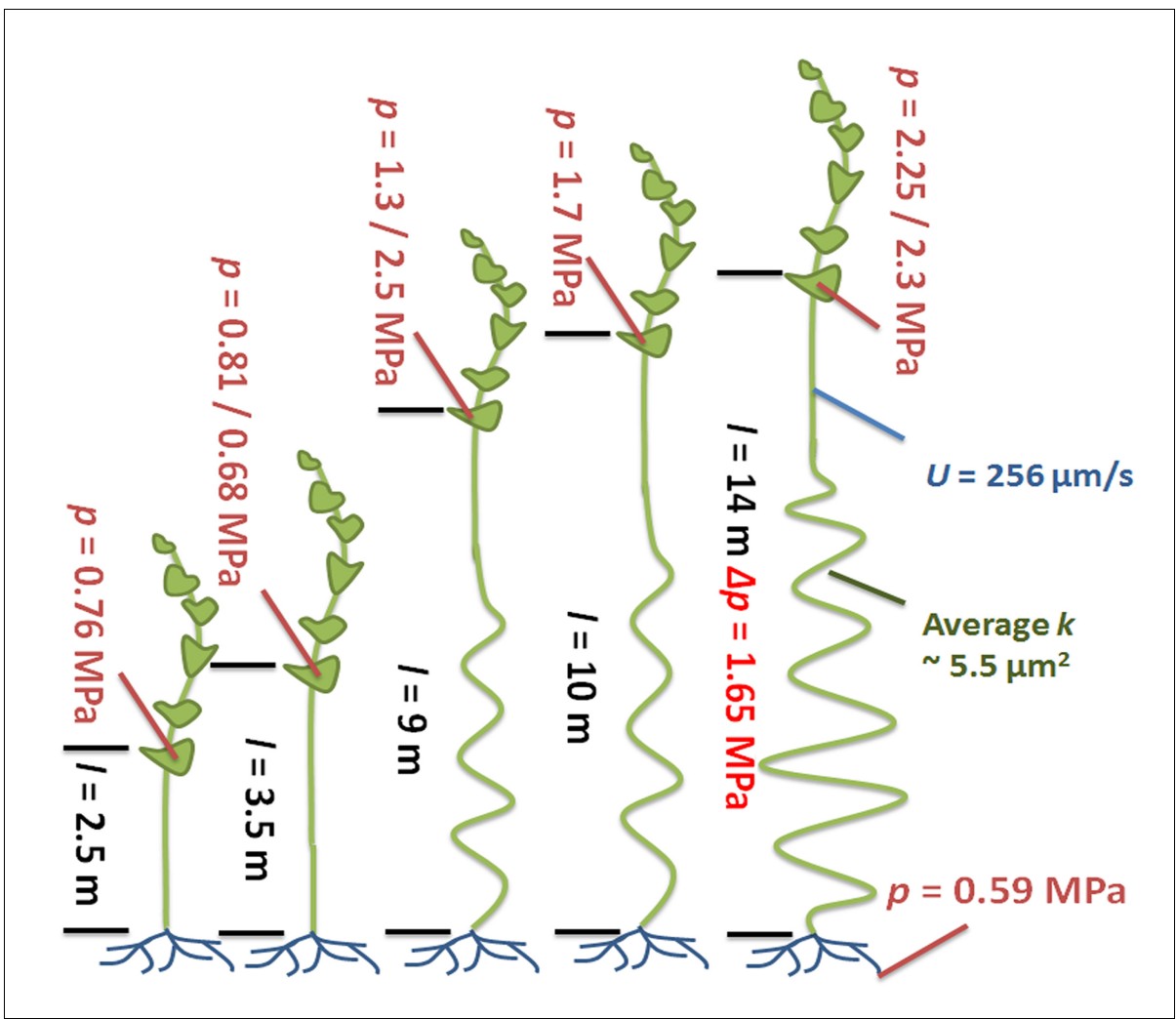

**Figure 4.** Phloem flow relevant parameters in a morning glory plant with increasing leafless stem length. An illustration summarizing the findings in large morning glory plants after artificial increase of source-to-sink transport distance achieved through continuous partial defoliation. Leaves were maintained only on the upper four meters of the plant and pressure was measured throughout the growth period. Plants with a short distance between the leaves and the roots maintain a relatively low sieve tube turgor pressure (crimson) in the source phloem in the range of 0.7–0.8 MPa, but the pressure scales with increasing length (black) of defoliated stem and the conductivity increases ~5 fold (green) compared with the 7.5 m long plant shown in *Figure 1* and *2*. Flow velocity (blue) was measured by $^{11}C$ application.

The following figure supplements are available for figure 4:

**Figure supplement 1.** In situ measurements on large morning glory plants.

**Figure supplement 2.** Morning glory stem growth.

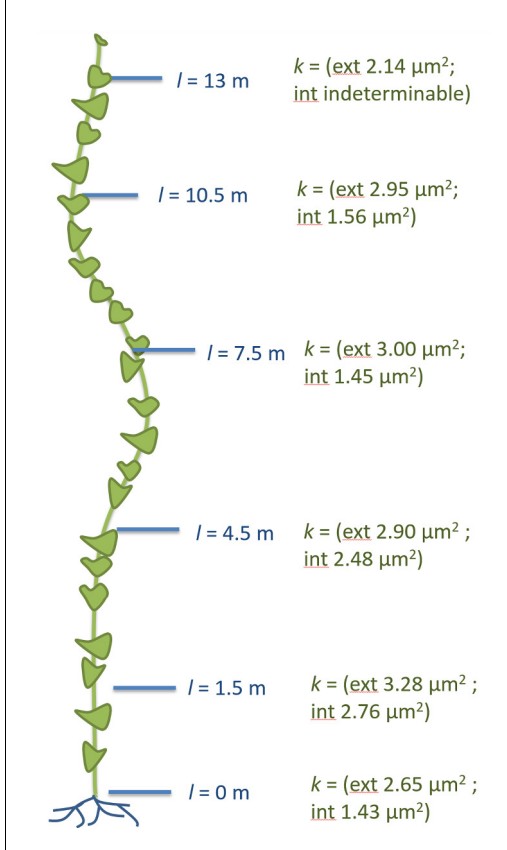

$l = 13\ m$ $k = (\text{ext } 2.14\ \mu m^2;$ int indeterminable)

$l = 10.5\ m$ $k = (\text{ext } 2.95\ \mu m^2;$ int $1.56\ \mu m^2)$

$l = 7.5\ m$ $k = (\text{ext } 3.00\ \mu m^2;$ int $1.45\ \mu m^2)$

$l = 4.5\ m$ $k = (\text{ext } 2.90\ \mu m^2 ;$ int $2.48\ \mu m^2)$

$l = 1.5\ m$ $k = (\text{ext } 3.28\ \mu m^2 ;$ int $2.76\ \mu m^2)$

$l = 0\ m$ $k = (\text{ext } 2.65\ \mu m^2 ;$ int $1.43\ \mu m^2)$

**Figure 5.** Anatomical adjustments to growth conditions. An illustration summarizing the findings in a large morning glory plant with leaves along the length of the stem. In contrast to partially defoliated plants, the conductivity remains relatively low, likely due to the shorter distance from source leaves to sink tissue.

The following source data is available for figure 5:

**Source data 1.** Source data of sieve tube geometrical parameters for **Figure 5**.

factor of 1.25 to account for helical stem growth (**Figure 4—figure supplement 2**). Despite the high pressure generated in source tissue, the pressure differential (assuming sink turgor of 0.59 MPa) would have allowed photoassimilates to be transported only 3.7 m at a measured flow velocity of 256 µm/s, if the tube geometry in the long plants was the same as in the smaller plants.

Instead, sieve tube conductivity increased 5-to-6 fold compared to fully foliated young plants (**Figure 3F**) mainly due to a significant increase in average sieve plate pore radius (**Figure 3E**, **Figure 3—figure supplement 3**). Taking into account the much higher conductivity, the estimated maximum transport distance is 20.35 m, which is in good agreement with the actual stem length, especially given that transport distance in the roots is not included. The change in sieve tube conductivity appears to be largely a response to the increased distance between source leaves and the sink (roots) as the average sieve tube conductivity of a 13 m long morning glory plant with source leaves all along its stem was only 2.8 µm$^2$ (**Figure 5**). Tilting of the sieve plate, and formation of compound instead of simple sieve plates increases the sieve plate area to allow larger pores (**Figure 3—figure supplement 2**). Measurement of the total phloem area from confocal micrographs (**Figure 3A**; **Figure 3—figure supplement 1**, **Figure 3—figure supplement 3**) indicated a significant increase of the external phloem area compared to plants with leaves along their entire stem, suggesting that the continuous extension of the distance between source and sink induces cambial activity to produce new sieve tubes with higher conductivity. Interestingly, despite a slight increase in tube conductivity, the conducting area of the internal phloem actually decreases (**Figure 3—figure supplement 3F**). This is attributed to the restricted space when secondary xylem and phloem is formed by the internal cambium. The significant reduction of the pith diameter in older tissue indicates the space limitations for internal secondary phloem. Therefore, the impact of the internal phloem on whole phloem transport decreases with increasing secondary growth.

These studies on morning glory show that when the transport length is increased, sieve tube conductivity and pressure increase accordingly. The observations provide strong support for pressure driven long distance transport.

## Discussion

In this study we address the major question about phloem transport in angiosperms: do phloem parameters scale in accord with the Münch hypothesis? We found that large morning glory vines exhibited a significant increase in the conductivity of sieve tubes in the external phloem and that this increase was primarily due to larger sieve plate pores. In addition to changes in sieve tube structure, the pressure in sieve tubes scaled with the distance between leaves and roots, such that morning

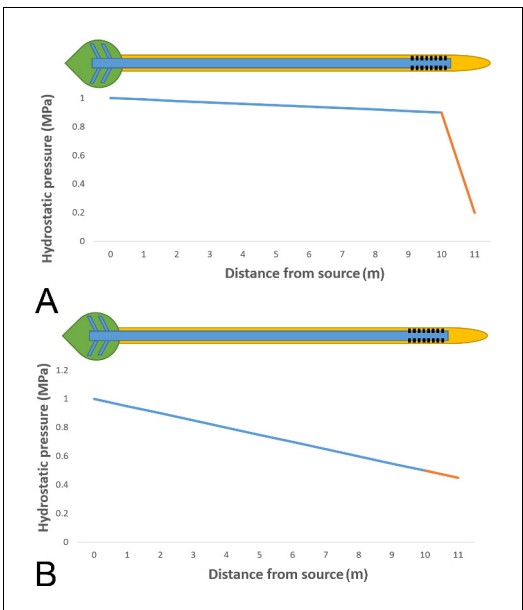

**Figure 6.** Phloem pressure gradients in relation to phloem unloading. Schematic drawing of a single source leaf (green) loading assimilates into the phloem (blue) and unloading through plasmodesmata (black) into a single root tip (yellow). (**A**) Independent of the plant size the pressure manifold model proposes nearly uniform high pressure along the stem, but large differences between sieve elements and surrounding cells in the unloading zone where the steepest gradients are found. (**B**) Our data suggest that tube resistance in the stem of morning glory plants consumes most of the pressure gradient, contrary to the high pressure manifold model.

glory plants with greater distance separating leaves and roots had significantly higher source turgor pressures. In summary, plants actively adjust flow relevant parameters to accommodate a passive pressure driven mass flow.

After almost a century, we finally have experimental evidence that addresses the key challenge to the pressure flow hypothesis. Although many cellular aspects, such as the function of most p-proteins and sieve element plastids, remains unclear, the overall transport concept proposed by Münch is supported. Prior to our study, many of the parameters governing phloem transport were poorly quantified. Thus, an additional important outcome of the study is a toolkit' for determining all of the parameters affecting flow. This means, that our work opens the door to new studies of phloem function. Foremost among the questions to be answered are how plants control the distribution of resources needed to support assimilate storage and the growth of new tissues.

Phloem unloading fills the reservoirs of the most important human food sources (primary sources such as fruits, roots, tubers, cereals, and through consumption also other sources such as meat) but our understanding of the unloading mechanism is similarly rudimentary. A currently popular modification of the Münch hypothesis that also accounts for phloem unloading is the high pressure manifold model (*Patrick, 2013*). This model proposes maintenance of high hydrostatic pressures throughout the sieve tube system with minimal pressure differences from source to sink, but high differences between sieve elements and surrounding cells in the unloading zone (*Figure 6A*). Our data, however, suggest that tube resistance will consume most of the energy provided by the source pressure to allow flow at the measured velocities (*Figure 6B*). Most importantly and contrary to the high pressure manifold model, small morning glory plants generate only low pressures sufficient for flow, but not additional large margins to maintain high pressure differentials in the unloading zone. This suggests that at least in the species investigated, either plasmodesmal conductance in the unloading zone is very high for passive unloading, or additional energy, e.g. in form of active unloading processes, is required. A better understanding of short and long distance flow patterns and assimilate distribution in plants will be necessary to resolve these important questions.

Organisms are structures whose integrity depends on active and passive flows. While active flows are dominated by proteins in the form of cytoskeletal elements or membrane transporters, passive flow is dependent on cell geometry and cellular connections in orchestration with concentration gradients and osmotic potentials (which are again generated by active flows). The lack of precise geometrical data of cellular connections is therefore a major problem, as this information forms the basis for modeling long distance flow patterns of assimilates, signaling molecules, and other substances. Large-scale efforts to collect cell geometrical data (a field one may call quantitative anatomy or 'Anatomics') at a resolution sufficient to extract sieve tube connections, diameters, length as well as plasmodesmal frequencies etc. throughout the plant are needed. Integration of physiological and molecular data (e.g. cell surface specific distribution of membrane transporters etc.) into a 'virtual model plant could lead to a major step forward in rationally designing crop plants to meet future

societal and environmental needs. To accomplish this methods for tissue preparation, automated acquisition, and reconstruction need to be combined, but the technology is available.

## Materials and methods

### Plant growth conditions

Due to the long growth period needed to attain large stem axes, plants were started in the greenhouse in February in pots at 23°C, with 60 to 70% relative humidity, and a 14/10 hr light/dark period (daylight plus additional lamp light [model #PL 90 (PL Lighting Systems, Ontario, Canada)] with a minimum irradiance of 150 µE m$^{-2}$ s$^{-1}$. The plants were moved outside at the end of May. The plants were grown on ropes to allow climbing and easy transport. Morning glory is a fast growing vine, but highly susceptible to pests. Primarily spider mites and aphids quickly grow in high numbers and have devastating impacts on the plants. In order to prevent potential effects of systemic pesticides on the plant's physiology, we chose to remove pests manually. Usually every day, but at least 4 days per week we checked every leaf and the stem of each plant and removed pests by spraying them off with a fine mist of water. All developing side branches and flower buds were pruned at the same time to maintain one single stem. Due to higher abundance of predators, pest infestation was much lower outside compared to in the greenhouse.

### Source sieve tube turgor measurements

Plants were transferred from the growth site to the lab just before the experiment was conducted. Phloem transport has been shown to be very stable and that diurnal changes have little or no impact on phloem transport velocities, likely due to sufficient starch pools even when photosynthesis capacity changes (*Windt et al., 2006*). Therefore we did not expect negative effects of movement of the plants. At the main vein of the lowermost source leaf, a small paradermal section to remove cortical tissue and to expose sieve elements was made with a fresh razor blade by hand. The section was immediately covered with phloem recovery medium consisting of 10 mM KCl, 10 mM NaCl, and 0.2 mM EDTA in unbuffered distilled water. Since the sieve tube to be measured must be the uppermost uninjured cell, the section was checked under the microscope. If sieve tube damage was observed, i.e. the cut was too deep, the leaf was removed and the plant was brought back to the growth site and a new attempt was performed after a recovery period of at least one day. If the section appeared useful in that the sieve tubes were well visible and appeared uninjured, a phloem mobile dye (CDFA or Esculin, Sigma-Aldrich, St. Louis, MO) was injected locally into the leaf apoplast about 5 cm upstream of the preparation site by pushing the dye solution through the stomata with the blunt end of a 3 ml syringe. About 2 cm$^2$ were filled with dye to provide sufficient fluorochromes for visualization after phloem loading. After filling up the apoplast, the phloem at the preparation site was observed for arrival of the fluorochromes which usually took 20–30 min. Translocating, intact tubes exposed to the surface were measured by impaling pico-gauges into the tube and monitoring the compression of the pico-gauge filling oil. Pico gauge production and pressure calculations were performed as described in detail in (*Knoblauch et al., 2014*).

### Sink root turgor pressure measurements

Sink root pressure was measured in root tips of morning glory plants grown in the greenhouse under standard conditions (see above). The plant was removed from its pot with the soil still attached to the roots and transferred into a plastic bag to keep roots moist. A small hole was cut into the bag, a single root was pulled out and covered with moist paper towels to maintain humidity. The root tip was immersed in water, and held in place by using a glass slide as weight. Measurements were carried out using pico gauges, produced by method A as described in (*Knoblauch et al., 2014*). About 2–3 min were given for temperature adjustments of the pico gauges to the surroundings. A Sutter Instruments (Novato, CA) model MPC 200 micromanipulator was used to control movement of the pico gauges. To take measurements, cortex cells of the root were impaled with a pico gauge and measurements were recorded at approximately 6 frames per second using a Leica DFC 450 camera, mounted to a Leica DM LFSA microscope equipped with a HCX Plan APO 40x lens. (Leica, Wetzlar, Germany). Data processing was conducted as described earlier (*Knoblauch et al., 2014*).

## Viscosity measurements

In situ viscosity measurements were conducted by loading 2-NBDG into source tissue of intact morning glory plants by pressure injection of 2-NBDG solution through the stomata using the blunt end opening of a 5 ml syringe. Fluorescence was observed downstream of the application site (*Knoblauch and van Bel, 1998*). Measurements were carried out on a Leica SP8 SMD white light laser system equipped with a pico quant FLIM attachment. The dye was excited with the 470 nm line of the supercontinuum laser and emission light was collected between 490 nm and 550 nm.

## Flow velocity measurements by carbon-11 labeled photoassimilate

Carbon-11, as $^{11}CO_2$, was generated by the $^{14}N(p,\alpha)^{11}C$ nuclear transformation (*Ferrieri and Wolf, 1983*), on an 19 MeV cyclotron (EBCO, Richmond, British Columbia, Canada). The $^{11}C$ was administered to a single leaf as $^{11}CO_2$ gas as a 30 sec pulse in continuously streaming air in a leaf cuvette with PAR 750 µmol m$^{-2}$ s$^{-1}$, as previously described (*Babst et al., 2013*). Leaf fixation, carbon export from the leaf, and $^{11}C$-photoassimilate transport velocity were monitored in real-time using a detector built into the leaf cuvette, and two detectors shielded with collimated lead and positioned to detect radioactivity from the stem. After the pulse of $^{11}CO_2$, plants were incubated in place with continuous airflow through the leaf cuvette for 2–3 hr to allow sufficient time for transport of $^{11}C$-photoassimilate through the stem. Transport velocity was calculated as the distance between the stem detectors divided by the transit time of the $^{11}C$ radioactivity between the first and second stem detectors. Positron Emission Tomography (PET) imaging was performed by positioning the stem and petiole within, and the load leaf outside of, the field of view of a microPET R4 (Concorde Microsystems, Knoxville, TN, USA) approximately 2 hr after $^{11}CO_2$ was administered. Data was acquired from stem and petiole radioactivity emissions for 10 min, and a 3-dimensional image was reconstructed as a single frame using microPET Manager 2.3.3.0 software (Siemens Medical Solutions USA Molecular Imaging; Knoxville, TN, USA).

## Collection of cell geometrical data

To obtain cell geometrical data for the number of sieve plate pores per plate, the radius of pores, the radius of the tube, and the thickness of the plate, morning glory stem segments of approximately 5 cm length were cut and immediately immersed into 70% ethanol at –20°C to prevent callose formation on the sieve plate. The tissue was freeze substituted for at least 1 week. After that, longitudinal- and cross-sections of approximately 1 mm length were taken and immersed in a medium containing 0.5% proteinase K, 8% triton X 100 at pH 8 (*Mullendore et al., 2010*). The tissue was digested at 55°C for 2–6 weeks with weekly changes of the digestion medium to remove the symplast. After digestion of the symplast the tissue was freeze dried and sputter coated. 40 SEM images per sample point were taken with a FEI (Hillsboro, Or) Quanta 200 FEG SEM to measure and quantify cell geometrical data. Measurements were performed with imageJ software as described in (*Mullendore et al., 2010*).

To collect data on the average length of sieve elements, longitudinal hand sections of freeze substituted tissue were taken with a razor blade, stained with calcofluor white and aniline blue, and images were taken with a Leica SP8 confocal microscope at excitation wavelength 405 nm and emission collection at 420–480 nm for calcofluor white, and 550–600 nm for aniline blue fluorescence. Measurements were performed with Leica LAS software.

## Evaluation of the mode of phloem unloading in the root tip

Primary leaves in morning glory seedlings were loaded with Carboxyfluorescein diacetate as described previously (*Wright and Oparka, 1997*). After 1 hr seedlings were removed from the pots, the soil was washed off the primary root and fluorescence was monitored with a I3 filter block on a Leica DMLFSA microscope and images were taken with a DFC-300 camera.

## Acknowledgements

We acknowledge support from the Franceschi Microscopy and Imaging Center at WSU. This work was supported by National Science Foundation grants IOS-1146500 (MK), IOS-1022106 and IOS-1456682 (NMH, MK), a Harvard Bullard Fellowship (MK), a Carlsberg Foundation grant

2013_01_0449 (KHJ), the USDA National Institute of Food and Agriculture, McIntire Stennis project 1009319 (BAB), and the U.S. Department of Energy, Office of Science, Office of Biological and Environmental Research, under contract number DE-AC02-98CH10886 (BAB).

---

# Additional information

## Funding

| Funder | Grant reference number | Author |
|---|---|---|
| Harvard Bullard Fellowship | | Michael Knoblauch |
| National Science Foundation | IOS-1022106 | Michael Knoblauch<br>Noel Michele Holbrook |
| National Science Foundation | IOS-1146500 | Michael Knoblauch |
| National Science Foundation | IOS-1456682 | Michael Knoblauch<br>Noel Michele Holbrook |
| U.S. Department of Energy | DE-AC02-98CH10886 | Benjamin A Babst |
| Carlsbergfondet | 2013_01_0449 | Kaare H Jensen |

The funders had no role in study design, data collection and interpretation, or the decision to submit the work for publication.

## Author contributions

MK, Designed the overall experimental strategy, Performed cell geometrical data, collection of samples, images, data processing, and analysis, Performed pressure measurements, Viscosity measurements, Wrote the manuscript; JK, Performed cell geometrical data, collection of samples, images, data processing, and analysis, Performed pressure measurements, Drafting or revising the article; DLM, Performed cell geometrical data, collection of samples, images, data processing, and analysis, Viscosity measurements; JAS, Designed the overall experimental strategy, Performed cell geometrical data, collection of samples, images, data processing, and analysis, Drafting or revising the article; BAB, Flow velocity measurements, Analysis and interpretation of data; SDB, Viscosity measurements, Analysis and interpretation of data; ACD, Performed cell geometrical data, collection of samples, images, data processing, and analysis; KHJ, NMH, Designed the overall experimental strategy, Performed cell geometrical data, collection of samples, images, data processing, and analysis, Wrote the manuscript

## Author ORCIDs

Michael Knoblauch, http://orcid.org/0000-0003-0391-9891

---

# Additional files

## Major datasets

The following dataset was generated:

| Author(s) | Year | Dataset title | Dataset URL | Database, license, and accessibility information |
|---|---|---|---|---|
| Michael Knoblauch, Jan Knoblauch, Daniel L Mullendore, Jessica A Savage, Benjamin A Babst, Sierra D Beecher, Adam C Dodgen, Kaare H Jensen, N Michele Holbrook | 2016 | Data from: Testing the Münch hypothesis of long distance phloem transport in plants | http://dx.doi.org/10.5061/dryad.n1g46 | Available at Dryad Digital Repository under a CC0 Public Domain Dedication |

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

## Appendix

### Evaluation of the conductivity k

The conductivity $k$ is required to compute the value of the pressure differential necessary to drive flow from equation (1). For the simple case of a cylindrical pipe, the conductivity is given by $k = r^2/8$, where $r$ is the radius of the pipe. However, the phloem transport pathway comprises a file of many short sieve elements lying end-to-end, separated by perforated screens (sieve plates) that each add considerable hydraulic resistance.

To account for the presence of sieve plates, we modeled the dependence of the conductivity $k$ on anatomical parameters following (**Jensen et al., 2014**, **2012**). Direct numerical simulations of low-Reynolds-number-flow through sieve plates have confirmed this analytical method (**Jensen et al., 2012**).

To compute the conductivity, we first take the hydraulic resistance of one cell to be the sum of contributions due to the cell lumen and sieve plate

$$R_1 = R_L + R_P. \tag{A1}$$

The lumen is approximated by a cylindrical tube of radius $r$ and length $\ell$. This yields a resistance of

$$R_L = 8\eta\ell/(\pi r^4), \tag{A2}$$

which is directly proportional to the sap viscosity $\eta$, so a higher sugar concentration leads to greater resistance. The sieve plate resistance $R_p$ takes into account the contribution from each pore arranged in parallel $R_p = \left(\sum R_{p,i}^{-1}\right)^{-1}$, where $R_{p,i}$ is the resistance of the i'th pore, given by

$$R_{p,i} = \frac{8\eta l_{p,i}}{\pi r_{p,i}^4} + \frac{3\eta}{r_{p,i}^3}. \tag{A3}$$

Here, $l_{p,i}$ is the length of each pore (the sieve plate thickness), while $r_{p,i}$ is the radius. The first term in **Eq. (A3)** corresponds to the Hagen-Poiseuille resistance to flow through the sieve pore, while the second term gives the resistance in the pore entrance region. To compute the total sieve plate resistance, it is sufficient to determine the mean pore radius $\bar{r}_p$ and the standard deviation $\sigma$ of the pore size distribution. Because the sieve pore radii follow a normal distribution, the plate resistance is well approximated by (Jensen et al., 2014)

$$R_p = \frac{3\eta}{\bar{r}_p^3} \frac{1}{N} \frac{1}{I(\alpha, \beta)}, \tag{A4}$$

where $N$ is the number of pores $I$ is a function of the non-dimensional parameters $\alpha = 8\ell_p/(3\pi\bar{r}_p)$ and $\beta = \sigma/\bar{r}_p$:

$$I(\alpha,\beta) = \left(\frac{1}{1+3\beta^2} + \frac{\alpha}{1+6\beta^2+3\beta^4}\right)^{-1}. \tag{A5}$$

The total resistance of a stem section of length $L$ is $R_{tot} = L/\ell(R_L + R_p)$, where the factor $L/\ell$ is the number of tubes arranged in series.

To compute the conductivity $k$, we finally use the relationship between $k$ and the total hydraulic resistance: $k = \eta L/(\pi r^2 R_{tot})$. This gives

$$k = \left[ \frac{8}{r^2} + \frac{3\pi}{N} \frac{r^2}{\bar{r}_p^3 \ell} \left( \frac{1}{1+3\beta^2} + \frac{\alpha}{1+6\beta^2+3\beta^4} \right) \right]^{-1}. \tag{A6}$$

