## [Decision Letter]

Thank you for submitting your article "How Plants Alter Sieve Tube Physiology and Geometry to Maintain Phloem Transport to Sinks" for consideration by *eLife*. Your article has been reviewed by three peer reviewers, and the evaluation has been overseen by a Reviewing Editor and Detlef Weigel as the Senior Editor.

The reviewers have discussed the reviews with one another and the Reviewing Editor has drafted this decision to help you prepare a revised submission.

Overall, the reviewers were positive and they agree that it is an important contribution to the field. There was, however, also a consensus that compared to the morning glory part, the oak part is preliminary and rather derivative. We therefore would like to ask you the following:

1) To remove the oak section from your manuscript, as well as the section on p-protein.

2) To concentrate on the presentation of your strong data set from morning glory, and polish the presentation to make your story more accessible to the readers.

Please find more details in the individual reviewer comments below (please note: they include the comments on the oak tree section for your feedback, but you can of course disregard them for the revision):

Reviewer #1:

Knoblauch and colleagues investigate two issues that challenged the Münch hypothesis of solute long distance transport in the phloem. 1. Structural components of the phloem and 2. The hydraulic conductivity of the phloem. I think this paper fills a big gap in our current understanding of phloem transport. The Münch hypothesis has been accepted for a very long time but the experimental fundaments were rather weak. The authors have done an outstanding and obviously very labor-intensive study to show that the Münch hypothesis can indeed explain how phloem transport happens. In terms of its significance there's no doubt in my mind that this paper will become *the* paper on this topic. Additionally, this paper raises several important issues for future research – one consequence is for example that phloem unloading must be re-evaluated. Nevertheless, I have some issues that I'd like to address.

1) I found this paper complicated to read and I believe the authors would help their case if they helped the reader a little bit more.

2) In the second paragraph of the subsection “Is osmotically driven pressure flow sufficient to account for phloem transport in tall trees and long vines?”, the authors state how many cells, sieve plates and so on were measured. This is certainly important – in my opinion it would help to give an estimate how many of these diameters or plate/pore numbers were observed for each individual “n" they were used for (Figure 1 and Figure 3 and Figure 6(?)).

3) Turgor measurements in root cortical cells were performed for 7.5 m plants. This value is also assumed for the defoliated plants and the oak trees. I appreciate that these measurements can't easily be done on oak roots but I wonder why this parameter wasn't obtained for the defoliated plants. I realize obtaining this parameter can't be done easily now so I would like to have an explanation why the 0.59 MPa can be assumed in both cases where the value wasn't determined. This is – in my opinion – particularly important in the case of the oak tree.

4) The section on trees is very interesting and the calculations sound – but in its current shape it is the weakest section of the paper and a bit iffy. It is stated that "Tube geometrical data provided evidence that tube conductivity in the trunk exceeds this [the conductivity of at least 10 µm2] by far" (subsection “Is osmotically driven pressure flow sufficient to account for phloem transport in tall trees and long vines?”, last paragraph). I assume these geometrical data were taken from the "bark samples" collected along the stem and branches (in the eleventh paragraph of the aforementioned subsection). If so: are these data part of the numbers given in the second paragraph? In any case it would be interesting to actually see these data! Without these data the only "hard" data on trees are the measurements of source mesophyll turgor and the height of the tree. In this context it is particularly important to convincingly make the case why the data from De Schepper et al. 2013, collected on a different oak species of a different age, can be used. The same goes for the sap viscosity and sink turgor that is assumed to be identical to that determined in morning glory.

5) Figure 1 and Figure 3: It is important to compare the values shown in both figures directly. For this purpose it would be extremely helpful if the panels appeared in the same order in both figures – this goes also for the scaling on the ordinate. Maybe also a Table for direct comparison would be a good idea. Also: why are there no standard errors/deviations shown in Figure 3?

Reviewer #2:

This manuscript describes an integrated series of structural, physical and physiological studies, conducted on Ipomoea nil (morning glory) and Quercus rubra (red oak), to address the question as to whether the Muench pressure flow hypothesis is valid. Collection of anatomical and physiological data was carried out on morning glory plants that had undergone two types of pruning. In one, branches and flowers were removed to produce ~ 7 meter long vines having source leaves distributed along the vine axis, but simplified sinks, namely the vegetative apex (and a few small expanding leaves) and the root system. In the other, as the vine grew, lower mature source leaves were removed to provide a situation in which the lower stem region was leafless; the same two sinks were in place. A pressure probe was used to determine the turgor pressure in the sieve elements (SE) located in the main vein of source leaves located along the plant axis. In parallel, anatomical studies performed on SE at various sites along the stem yielded parameters used to compute sieve tube radius, sieve plate pore numbers and radii, SE lengths and local phloem conductivity. These data were then used to calculate the theoretical pressure gradient that would be required to achieve the measured flow velocities. Based on data presented, the authors conclude that their study on morning glory provides experimental support for the phloem pressure flow model.

1) The techniques employed in this study are quite elegant and, if the manuscript had been well written (and proofed carefully by all authors) it could have been a pleasure to read. In addition, it is disappointing that, having gone to such pains to collect the anatomical data, the authors used assumptions from plants like *Arabidopsis* to design aspects of their experiments. For example, they assumed that, as in *Arabidopsis*, symplasmic phloem unloading occurs in morning glory roots. Added to this, even if symplasmic unloading were to take place, making turgor pressure measurements on cortical cells, rather than on SEs located in the sink region of the root, confounds data interpretation. Given that numerous cells are positioned between the SE and the cortical cell layer, the measured p value of 0.59 MPa will be lower than that within the root sieve tube system. Putting this aside, one has to puzzle as to how the authors arrived at the value of 0.485 MPa for the "measured pressure differential." Inspection of the data presented in Figure 2 indicates an average p value of 1.2 MPa for the lower leaves which can be applied to the bottom leaf; this gives a delta p = 0.6 MPa, and a distance of ~2.5 m (measured based on the figure) from this bottom leaf and the point where the p root value was measured. This gives a delta p of 0.24 MPa, which we know would be an overestimate. So, the data are in the ballpark, but the authors have not yet knocked the ball out of the park!

2) Results from the second type of pruned/manipulated vine are of more interest as they uncover a nice interactive network at play to adjust the parameters of the phloem, in a positional-dependent manner, in response to the imposition of increasing distance between the root and the source leaves.

3) The work on red oak is greatly weakened by being based on just too many assumptions and so we recommend that it be removed from the manuscript. The same can be said of the p-protein aspect of the manuscript.

4) Discussion of the pressure manifold model is weak; the data presented in Figure 2 provides support for this model. However, the data in Figure 4 suggests that for plants like trees, the manifold model might need to be revised.

Reviewer #3:

The authors aim to address two fundamental questions regarding pressure flow hypothesis that have been long debated and that are key to understand phloem physiology: the continuity of the flow (are the pores blocked by p-proteins?) and conductivity in trees (is the pressure as high as necessary for the increased distances?).

They argue that part of the controversy is caused by lack of reliable data due to technical constrains and develop new methods to avoid these problems. Here they measured *in situ* sap viscosity and phloem pressure in a morning glory plant partly defoliated and also the red oak tree to show its phloem characteristics, by great increase of sieve tubes conductivity, is still in accordance with the Munch model, a matter of debate for long distance sap flow movement in trees.

They also provide evidence in *Arabidopsis* that p-proteins, long thought to block phloem sap flow at sieve plates, are in fact able to diffuse through them and thus still allow pressure driven sap flow.

The authors, using a set of complementary approaches, are able here to apprehend the different parameters driving the sap flow in the sieve tube. Their results validate the Munch pressure flow model in sieve tubes and show the plasticity of phloem to adapt its physical characteristics through modified source/sink balance. The techniques developed by this team are a great technical advance for phloem studies.

The data is clear, almost everywhere well explained and in accordance with the authors' assumptions. Nevertheless, several points should be addressed:

1) The authors don't explain how their FLIM calibrations confirm the efficient use of 2-NBDG as a molecular rotor suitable for sap viscosity assessment. This point need further clarification and may require a supplemental figure. Also, the authors argue for that the viscosity value obtained through the FLIM experiment corresponds to an 18% sucrose concentration, in accordance to published values of various plant species. It would have been better to confirm this result directly in the morning glory vine model to strengthen this result.

2) Measuring the phloem parameters in the red oak tree model, the authors assume a sap viscosity of 1.7 mPas, in accordance with the value obtained through FLIM experiments with the morning glory model. If FLIM experiment are not feasible in trees, why not try here to obtain an estimation of sap phloem viscosity through concentrations determination of extracted phloem sap contents (previously used technique to estimate sap viscosity mentioned by the authors in the text)? It would have more strength than just an assumption.

3) The authors use *Arabidopsis* plants expressing YFP-tagged SEOR1 to show in vivo p-protein agglomerations block sieve tubes only occasionally and temporarily, which doesn't affect phloem transport. But we still don't know if the situation of P-proteins is same or similar in other plant species. May I ask the authors to discuss?

---

## [Author Response]

We have followed all suggestions with a few exceptions that we have explained in the text. We would like to change the title of the manuscript from “How Plants Alter Sieve Tube Physiology and Geometry to Maintain Phloem Transport to Sinks” to “Testing the Münch hypothesis of long distance phloem transport in plants”, because we feel that this better reflects the work presented in the revised manuscript.

*Overall, the reviewers were positive and they agree that it is an important contribution to the field. There was, however, also a consensus that compared to the morning glory part, the oak part is preliminary and rather derivative. We therefore would like to ask you the following:*

1) To remove the oak section from your manuscript, as well as the section on p-protein.

Both sections have been removed.

2) To concentrate on the presentation of your strong data set from morning glory, and polish the presentation to make your story more accessible to the readers.

The revised manuscript includes only the morning glory data and the text has been revised to emphasize the significance of these data to the long-standing question of phloem transport in large and long plants.

*Please find more details in the individual reviewer comments below (please note: they include the comments on the oak tree section for your feedback, but you can of course disregard them for the revision):*

*Reviewer #1:*

*Knoblauch and colleagues investigate two issues that challenged the Münch hypothesis of solute long distance transport in the phloem. 1. Structural components of the phloem and 2. The hydraulic conductivity of the phloem. I think this paper fills a big gap in our current understanding of phloem transport. The Münch hypothesis has been accepted for a very long time but the experimental fundaments were rather weak. The authors have done an outstanding and obviously very labor-intensive study to show that the Münch hypothesis can indeed explain how phloem transport happens. In terms of its significance there's no doubt in my mind that this paper will become the paper on this topic. Additionally, this paper raises several important issues for future research – one consequence is for example that phloem unloading must be re-evaluated. Nevertheless, I have some issues that I'd like to address.*

1) I found this paper complicated to read and I believe the authors would help their case if they helped the reader a little bit more.

We believe that the removal of the P-protein data and oak data has made the story much less complex and easier to access.

*2) In the second paragraph of the subsection “Is osmotically driven pressure flow sufficient to account for phloem transport in tall trees and long vines?”, the authors state how many cells, sieve plates and so on were measured. This is certainly important – in my opinion it would help to give an estimate how many of these diameters or plate/pore numbers were observed for each individual “n" they were used for (Figure 1 and Figure 3 and Figure 6(?)).*

All sample sizes (n) were provided in the original manuscript in the figure legends as requested in the *eLife* author guide. However, we agree that it would be beneficial to provide an overview and to accomplish this we have added a sentence to the Introduction.

3) Turgor measurements in root cortical cells were performed for 7.5 m plants. This value is also assumed for the defoliated plants and the oak trees. I appreciate that these measurements can't easily be done on oak roots but I wonder why this parameter wasn't obtained for the defoliated plants. I realize obtaining this parameter can't be done easily now so I would like to have an explanation why the 0.59 MPa can be assumed in both cases where the value wasn't determined. This is – in my opinion – particularly important in the case of the oak tree.

Because the oak data have been removed from the manuscript, the central point of the concern is obsolete. However, we would like to explain why we did not measure root pressure in partly defoliated long plants. As described in the manuscript, measurement of root turgor pressure required the removal of the root system from the pot, putting the root system in a plastic bag, cutting a hole in the bag and pulling a root out, keeping the root moist and mounting it on the microscope stage. Doing this with a large plant appeared not feasible. A single minor crack in the highly delicate stems of morning glory would have jeopardized the project and likely have resulted in errors in the measured turgor pressures.

4) The section on trees is very interesting and the calculations sound – but in its current shape it is the weakest section of the paper and a bit iffy. It is stated that "Tube geometrical data provided evidence that tube conductivity in the trunk exceeds this [the conductivity of at least 10 µm2] by far" (subsection “Is osmotically driven pressure flow sufficient to account for phloem transport in tall trees and long vines?”, last paragraph). I assume these geometrical data were taken from the "bark samples" collected along the stem and branches (in the eleventh paragraph of the aforementioned subsection). If so: are these data part of the numbers given in the second paragraph? In any case it would be interesting to actually see these data! Without these data the only "hard" data on trees are the measurements of source mesophyll turgor and the height of the tree. In this context it is particularly important to convincingly make the case why the data from De Schepper et al. 2013, collected on a different oak species of a different age, can be used. The same goes for the sap viscosity and sink turgor that is assumed to be identical to that determined in morning glory.

This part has been removed from the manuscript.

5) Figure 1 and Figure 3: It is important to compare the values shown in both figures directly. For this purpose it would be extremely helpful if the panels appeared in the same order in both figures – this goes also for the scaling on the ordinate. Maybe also a Table for direct comparison would be a good idea. Also: why are there no standard errors/deviations shown in Figure 3?

We had provided the source data, but we agree that it would be beneficial to have an easy accessible direct comparison of the data between the plants. We therefore have generated Figure 3—figure supplement 3 showing a graphical comparison as well as tables for the individual parameters. We have also included the error bars in this figure supplement as they appeared distracting in Figure 3. A statement in Figure 3 legend refers readers to Figure 3—figure supplement 3 for standard deviations.

*Reviewer #2:*

*1) The techniques employed in this study are quite elegant and, if the manuscript had been well written (and proofed carefully by all authors) it could have been a pleasure to read. In addition, it is disappointing that, having gone to such pains to collect the anatomical data, the authors used assumptions from plants like Arabidopsis to design aspects of their experiments. For example, they assumed that, as in Arabidopsis, symplasmic phloem unloading occurs in morning glory roots.*

Not only in *Arabidopsis*, but in all plants studied so far including monocots, symplastic unloading has been shown in the root unloading zone. We have added Figure 2—figure supplement 3 showing proof of symplastic unloading in root tips in morning glory.

*Added to this, even if symplasmic unloading were to take place, making turgor pressure measurements on cortical cells, rather than on SEs located in the sink region of the root, confounds data interpretation.*

We would certainly have preferred to take direct sink sieve tube measurements in addition to cortical measurements, but as noted in the original manuscript, this is impossible. In roots the phloem is located in the central cylinder which would require splitting the root in half to access the sieve tubes for measurements. How this could be done without injury and major impacts on transport, unloading, and turgor is not clear to the authors.

*Given that numerous cells are positioned between the SE and the cortical cell layer, the measured p value of 0.59 MPa will be lower than that within the root sieve tube system.*

Correct. This supports our statement in the original manuscript. Our calculations show how much pressure is needed to overcome frictions within the tube system. The results show that most of the pressure differential will be consumed and that there is not a large margin for a high-pressure manifold system. Therefore, one has to assume that symplastic unloading does not require large pressure differentials as outlined in the Discussion and Figure 7. But the results do not contradict a pressure flow model.

*Putting this aside, one has to puzzle as to how the authors arrived at the value of 0.485 MPa for the "measured pressure differential." Inspection of the data presented in Figure 2 indicates an average p value of 1.2 MPa for the lower leaves which can be applied to the bottom leaf; this gives a delta p = 0.6 MPa, and a distance of ~2.5 m (measured based on the figure) from this bottom leaf and the point where the p root value was measured. This gives a delta p of 0.24 MPa, which we know would be an overestimate.*

Figure 2 shows 5 individual measurements, which average 1.08 MPa, not 1.2 MPa. Subtracting 0.59 MPa of sink turgor results in a pressure differential of 0.49 MPa. We do not see any problem with our calculations and the text clearly states what we have done. The only discrepancy is that we provided data with three digits (0.485MPa) instead of rounding to two digits (0.49 MPa). We have changed this in the revised manuscript.

So, the data are in the ballpark, but the authors have not yet knocked the ball out of the park!

According to our measurements, 0.21MPa are required to overcome friction and to drive flow at the measured velocities over a distance of 1m through the tube system. As concluded in the original manuscript, the measured pressure is high enough to drive the flow to any sink in the plant as the maximum source to sink distance does not exceed 2 m. Given the variability of tube geometry and the unknown numbers of plasmodesmata connections and size exclusion limits in the unloading zone it would be quite unexpected if the numbers matched 100% and it would be presumptuous for us to claim that we “knocked the ball out of the park” (and we actually do not see where we potentially have done this to elicit the reviewer’s comment). It is our opinion, however, that we are permitted to claim that we have provided strong support for pressure driven mass flow. Certainly our data put to rest the idea that pressure driven flow is not capable of transporting photoassimilates over long distances.

*2) Results from the second type of pruned/manipulated vine are of more interest as they uncover a nice interactive network at play to adjust the parameters of the phloem, in a positional-dependent manner, in response to the imposition of increasing distance between the root and the source leaves.*

3) The work on red oak is greatly weakened by being based on just too many assumptions and so we recommend that it be removed from the manuscript. The same can be said of the p-protein aspect of the manuscript.

This section has been removed from the manuscript.

4) Discussion of the pressure manifold model is weak; the data presented in Figure 2 provides support for this model. However, the data in Figure 4 suggests that for plants like trees, the manifold model might need to be revised.

We do not see why Figure 2 provides support for the high-pressure manifold model. The model requires significant pressures in the unloading zone as outlined in Patrick 2013, and it appears that reviewer 1 agrees with us that the conclusions from our data does not support this model. We would prefer to keep the conclusions as presented.

*Reviewer #3:*

*[…] The data is clear, almost everywhere well explained and in accordance with the authors' assumptions. Nevertheless, several points should be addressed:*

*1) The authors don't explain how their FLIM calibrations confirm the efficient use of 2-NBDG as a molecular rotor suitable for sap viscosity assessment. This point need further clarification and may require a supplemental figure.*

The calibration curve shown in Figure 2—figure supplement 1 is generated by measuring 2-NDBG lifetime versus known viscosities of aqueous sucrose solutions. We have added better explanation in the figure legend to clarify this.

Also, the authors argue for that the viscosity value obtained through the FLIM experiment corresponds to an 18% sucrose concentration, in accordance to published values of various plant species. It would have been better to confirm this result directly in the morning glory vine model to strengthen this result.

All phloem sap viscosity values (based on stylectomy or exudates) are currently based on estimations and are not measured in situ. The small volume in stylectomy leads to rapid concentration and viscosity changes because of evaporation which can be limited, but not entirely prevented. Exudates often contain contaminations from neighboring cells and the apoplast and oxidization may lead to gelling of the sap (e.g. in Cucurbits). In addition, sieve tube viscosity is dependent on all solutes in the sap, not only on sucrose. Since the primary aim was to measure viscosity and not sucrose concentrations, we decided to develop a method for in situ measurements by FLIM, calibrated against known viscosities which we believe provides better values than invasive methods.

2) Measuring the phloem parameters in the red oak tree model, the authors assume a sap viscosity of 1.7 mPas, in accordance with the value obtained through FLIM experiments with the morning glory model. If FLIM experiment are not feasible in trees, why not try here to obtain an estimation of sap phloem viscosity through concentrations determination of extracted phloem sap contents (previously used technique to estimate sap viscosity mentioned by the authors in the text)? It would have more strength than just an assumption.

This section has been removed from the manuscript.

3) The authors use Arabidopsis plants expressing YFP-tagged SEOR1 to show in vivo p-protein agglomerations block sieve tubes only occasionally and temporarily, which doesn't affect phloem transport. But we still don't know if the situation of P-proteins is same or similar in other plant species. May I ask the authors to discuss?

This section has been removed from the manuscript.